# Pregnancy outcomes of immigrant women living in Korea: A population-based study

**Geum Joon Cho[1], Ho Yeon Kim[1], Hyun Sun Ko[2], Hae Joong Cho[3], Seong Yeon Hong[4], Eunjin Noh[5], Young Ju Jeong [6]***

1 Department of Obstetrics and Gynecology, Korea University College of Medicine, Seoul, Republic of Korea, 2 Department of Obstetrics and Gynecology, Seoul St. Mary's Hospital, College of Medicine, The Catholic University of Korea, Seoul, Republic of Korea, 3 Department of Obstetrics & Gynecology, College of Medicine, WonKwang University, Iksan, Republic of Korea, 4 Department of Obstetrics and Gynecology, Catholic University of Daegu School of Medicine, Gyeongsan-si, Republic of Korea, 5 Korean University Guro Hospital Smart Healthcare Center, Seoul, Republic of Korea, 6 Department of Obstetrics and Gynecology, Research Institute of Clinical Medicine of Jeonbuk National University-Biomedical Research Institute of Jeonbuk National University Hospital, Jeonju, Korea

* yjjeong@jbnu.ac.kr

**Data Availability Statement:** The access to raw data of the Korean Health Insurance Review and Assessment (HIRA) service is regulated by the Rules for Data Exploration and Utilization of the HIRA. Data are available from the Health Insurance

## Abstract

Although there is a high rate of pregnant immigrant women in Korea, little is known regarding their pregnancy outcomes. The aim of this study was to evaluate the pregnancy outcomes of immigrant women in Korea. Data for all pregnant women who gave birth between January 1, 2007 and December 31, 2016 were obtained using the Health Insurance Review and Assessment Service Database. Pregnant women were divided into two groups: Korean and immigrant women. The main outcome measures were adverse pregnancy outcomes including gestational diabetes of mellitus, preeclampsia, cesarean section, placental abrnomalities, and postpartum hemorrhage. The odds of gestational diabetes mellitus, preeclampsia, cesarean section, placental previa, placental abruptio, and postpartum hemorrhage was compared between the two groups. Among 4,439,778 pregnant women who gave birth during the study period, 168,940 (3.8%) were immigrant women. The odds of gestational diabetes mellitus (adjusted OR: 1.24; 95% CI: 1.21, 1.28), and cesarean section (adjusted OR: 1.26; 95% CI: 1.25–1.28)were higher in immigrant women than in Korean women, but the odds of preeclampsia (adjusted OR: 0.84; 95% CI: 0.81–0.86) and postpartum hemorrhage (adjusted OR 0.96, 95% CI 0.94–0.97) was lower in immigrant women than in Korean women. Immigrant women had different pregnancy outcomes. Pregnancy and postpartum management that reflects these characteristics will be necessary for immigrant women.

## Introduction

South Korea has been suffered from low birth rate predominantly due to fast urbanization that the total fertility or birth rate is below 1 in 2021(Korea Statistical Information Service[KOSIS], 2021). Consequently, this low birth rate has further slowed the population growth and has rapidly transitioned to an aged society than any other country in the world. To overcome this

Review and Assessment Service database for researchers who meet the criteria for access to confidential data only after receiving an approval for the data use from the data access committee of the HIRA. HIRA data can be requested through its website (http://opendata.hira.or.kr).

**Funding:** This research was supported by the Korean society of Maternal Fetal Medicine Research Fund. (grant no. KSMSM-2017-001) The funders had no role in study design, data collection and analysis, decision to publish, or preparation of the manuscript.

**Competing interests:** The authors have declared that no competing interests exist.

**Abbreviations:** GDM, gestational diabetes; HIRA, Health Insurance Review and Assessment Service; KNHI, Korea National Health Insurance; UAE, uterine arterial embolization; CS, Cesarean section; PH, peripartum hysterectomy; PPH, postpartum hemorrhage; CCI, Charlson Comorbidity Index.

social phenomenon, the Korean government has implemented more favorable environment for childbearing and encouraged interracial marriages as a method of securing a labor force [1, 2] Therefore, interracial marriages in Korea are common, particularly in the past two decades due to "wife shortages" [3]. In 2018, 22,698 interracial marriages were registered, with 16,608 foreign wives and 6,090 foreign husbands. A total of 257,622 spouses were recorded; therefore, 8.81% of all marriages in Korea were interracial in 2018 [4].

After immigration various factors originating from different cultures, economy and communication may lead to problems with health outcomes of foreign wives [5]. In reproductive health, many obstetric investigations across different countries have limited adjustment for ethnicity, compared to native women, immigrant women have an increased risk of preterm birth, low birth weight and cesarean section [6–9]. In addition, it has been reported that immigrants in countries of resettlement have a greater risk of gestational diabetes mellitus (GDM) than women in receiving countries [10]. Thus, influxes of migrant women of childbearing age to receiving countries have made their perinatal health status a key priority for many governments [9].

Adverse pregnancy-related outcomes including hypertensive disorders, GDM, small for gestational age and postpartum hemorrhage have been steadily increasing especially due to advanced maternal age and higher body mass index(BMI) in South Korea [11–13]. Perinatal outcomes are influenced by not only maternal age and BMI but also smoking and alcohol consumption and further influences of racial and ethnic disparities have been documented. Evidences have shown increasing trends of interracial marriages in South Korea and this phenomenon may be related to adverse pregnancy outcomes which can lead to enormous burden on society.

We previously investigated birth outcomes using data obtained from the National Birth Registry of the Korean Statistical Office and found that the birthweight was lower, and that the risk of low birth weight was increased in foreign pregnant women compared to Korean pregnant women [14]. However, data obtained from the National Birth Registry of the Korean Statistical Office has a limitation in evaluating pregnancy outcomes, such as preeclampsia and GDM, due to lack of data. The aim of this study was to evaluate the pregnancy outcomes of immigrant women in Korea.

## Materials and methods

### Study design

Using the HIRA Database, we identified all pregnant women who gave birth between January 1, 2007 and December 31, 2016. Maternal ethnicity was identified and divided into two categories: Korean pregnant women and immigrant pregnant women.

### Data characteristics

The study data were collected from the Health Insurance Review and Assessment Service (HIRA) Database for 2007–2016. In Korea, 97% of the population is required to enroll in the Korea National Health Insurance (KNHI) program, with the exception of the remaining 3% of the population who are treated under the Medical Aid Program. Healthcare providers are required by health insurance policies to allow the HIRA to review medical costs. Thus, the HIRA database contains information on all claims for approximately 50 million Koreans, and nearly all information regarding the incidence of disease can be obtained from this centralized database, with the exception of procedures that are not covered by insurance, such as cosmetic surgery. According to the Act on the Protection of Personal Information Maintained by Public Agencies, the HIRA prepares the claims data by concealing individual identities. The HIRA

database we received included unidentifiable codes that represented individuals, together with age, diagnosis, and a list of prescribed procedures.

## Ethical consideration

This study was approved by the Institutional Review Committees of Korea University Guro Hospital (KUGH17273). The informed consent was waived by this IRB because the HIRA database does not include individual identities and because of retrospective nature of this study. This study comprises of third-party data therefore authors cannot share data nor legally distribute.

## Dataset and outcomes

Using the International Classification of Diseases, Tenth Revision (ICD-10), diagnoses of pre-eclampsia, GDM, placental previa, placental abruptio, and postpartum hemorrhage(PPH) were obtained from the KNHI Claims Database. Parity, and the rate of cesarean section, induction of labor vacuum delivery, uterine arterial embolization (UAE), and peripartum hysterectomy (PH) were also identified using the presence of a Korea Medical Insurance electronic data interchange (EDI) code.

Main outcome of this study were adverse pregnancy outcomes including GDM, preeclampsia, cesarean section, induction of labor, vacuum delivery, placenta previa, placenta abruption, PPH, peripartum hysterectomy and UAE. Age, parity and CCI were adjusted for multiple logistic regression analysis.

## Covariates

The Charlson Comorbidity Index (CCI) was identified using ICD-10 codes in order to adjust for pre-pregnancy factors. The CCI has been known to be a useful and most widely used tool to measure comorbid disease status including cardiovascular diseases, diabetes, malignancies and autoimmune diseases or casemix in health care databases that the higher the score the more comorbid conditions are present [15]. Acute myocardial infarction, congestive heart failure, peripheral vascular disease, cerebral vascular accident, dementia, pulmonary disease, connective tissue disorder, peptic ulcer, liver disease, diabetes, diabetes complications, paraplegia, renal disease, cancer, metastatic cancer, severe liver disease, and HIV were analyzed in this study. The CCI score was categorized as 0, 1, 2, 3, and $\geq$ 4 [16].

## Statistical analysis

The Student's t-test was used to compare continuous variables between Korean and immigrant pregnant women, while the chi-square test was used to compare categorical variables. A Secular trend in the rate of immigrant pregnant women was determined and compared across years using the χ2 Cochran-Armitage test. Multivariable logistic regression analysis was used to estimate the adjusted odds ratio (OR) and the 95% confidence intervals (CIs) for adverse pregnancy outcomes. The model was run stepwise. Statistical analyses were performed using SAS for Windows, version 9.4 (SAS Inc., Cary, NC, USA).

## Result

Among 4,439,778 pregnant women who gave birth during the study period, 168,940 (3.8%) were immigrant women. The rate of immigrant women increased from 2.98% in 2007 to 4.53% in 2016 (p-value for trend < 0.01) (Fig 1).

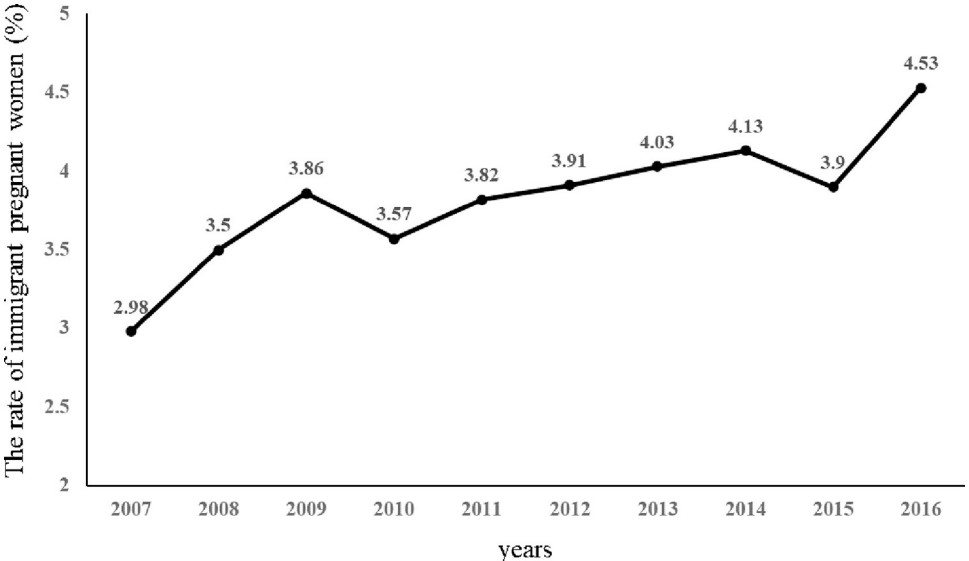

**Fig 1. Secular trends of the rate of immigrant pregnancy women in Korea.**

Table 1 shows the basic characteristics of the study population according to maternal ethnicity. Foreign women were younger and more primiparous than Korean women, and tended to have lower CCI score than Korean women.

Table 2 shows the pregnancy outcomes of the study population according to maternal ethnicity. Foreign women had a lower prevalence of preeclampsia than Korean women. The rate of cesarean section (CS), induction was lower in foreign women than Korean women. The incidence of placenta abruption, placenta previa, PPH, PH, and UAE were lower in immigrant women than Korean women.

Table 3 represents the odds of adverse pregnancy outcomes by maternal ethnicity. The odds of GDM, preeclampsia, CS, placental abruption, placental previa, PPH, PH, and UAE was lower in foreign women than Korean women. However, after adjustment for age, parity,

**Table 1. Basic characteristics of the study population.**

|  | Korean women (n = 4,270,838) | Immigrant women (n = 168,940) | P-value |
|---|---|---|---|
| Age (years) | 31.08 ± 4.07 | 27.20 ± 5.30 | < 0.01[a] |
| Primiparity (n, %) | 2,210,063 (51.75) | 111,787 (66.17) | < 0.01[b] |
| CCI score (n, %) |  |  | < 0.01[b] |
| 0 | 20,93,245 (49.01) | 129,849 (76.86) |  |
| 1 | 1,200,144 (28.10) | 28,843 (17.07) |  |
| 2 | 613,132 (14.36) | 7,557 (4.47) |  |
| 3 | 240,100 (5.62) | 1,906 (1.13) |  |
| ≥ 4 | 124,217 (2.91) | 785 (0.46) |  |

Values are expressed as mean (SD) or number, %.

a p-value associated with student t-test

b p-value associated with chi-square test

Abbreviations: CCI: Charlson Comorbidity Index

CCI included acute myocardial infarction, congestive heart failure, peripheral vascular disease, cerebral vascular accident, dementia, pulmonary disease, connective tissue disorder, peptic ulcer, liver disease, diabetes, diabetes complications, paraplegia, renal disease, cancer, metastatic cancer, severe liver disease, and HIV.

**Table 2. The pregnancy outcomes of the study population.**

|  | Korean women (n = 4,270,838) | Immigrant women (n = 168,940) | *P*-value |
|---|---|---|---|
| GDM (n, %) | 182,874 (4.28) | 5,526 (3.27) | < 0.01 |
| Preeclampsia (n, %) | 166,202 (3.89) | 4,871 (2.88) | < 0.01 |
| Cesarean section (n, %) | 1,613,028 (37.77) | 61,036 (36,13) | < 0.01 |
| Induction (n, %) | 995,220 (23.30) | 35,769 (21.17) | < 0.01 |
| Vacuum delivery (n, %) | 252,463 (5.91) | 13,373 (7.92) | < 0.01 |
| Placental previa (n, %) | 43,480 (1.02) | 1,248 (0.74) | < 0.01 |
| Placental abruption (n, %) | 15,232 (0.36) | 502 (0.30) | < 0.01 |
| PPH (n, %) | 376,800 (8.82) | 13,610 (8.06) | < 0.01 |
| Peripartum hysterectomy (n, %) | 4,734 (0.11) | 139 (0.08) | < 0.01 |
| UAE (n, %) | 6,126 (0.14) | 181 (0.11) | < 0.01 |

Values are expressed as number,%

*P*-value are all associated with chi-square test.

Abbreviations: GDM: Gestational diabetes mellitus, PPH: Postpartum hemorrhage, PH: Peripartum hysterectomy, UAE: Uterine arterial embolization

and CCI, the odds of GDM (adjusted OR: 1.24; 95% CI: 1.21, 1.28), and CS (adjusted OR: 1.26; 95% CI: 1.25–1.28), were higher in immigrant women than Korean women, but the odds of preeclampsia (adjusted OR: 0.84; 95% CI: 0.81–0.86) and PPH (adjusted OR: 0.96; 95% CI: 0.94–0.97) was lower. The odds of placental abruption (adjusted OR: 0.96; 95% CI: 0.88–1.05), PH (adjusted OR: 1.17; 95% CI: 0.98–1.38), and UAE (adjusted OR: 1.07; 95% CI: 0.92–1.24) was not different between the two groups after adjustment for age, parity, and CCI.

## Discussion

In this study, we evaluated pregnancy outcomes according to maternal ethnicity and found differences in pregnancy outcomes. Immigrant women were protected against having preeclampsia and PPH and higher risk of GDM, and CS compared to Korean women after adjustment for age, parity, and CCI.

**Table 3. Odds ratio and 95% confidence intervals of adverse pregnancy outcomes of 155 immigrant women.**

|  | Unadjusted ORs (95% CI) | Model 1 | Model 2 | Model 3 |
|---|---|---|---|---|
| GDM | 0.76 (0.74, 0.78) | 1.06 (1.04, 1.09) | 1.06 (1.03, 1.09) | 1.24 (1.21, 1.28) |
| Preeclampsia | 0.73 (0.71, 0.76) | 0.80 (0.78, 0.83) | 0.80 (0.77, 0.82) | 0.84 (0.81, 0.86) |
| Cesarean section | 0.93 (0.92, 0.94) | 1.23 (1.22, 1.24) | 1.23 (1.21, 1.24) | 1.26 (1.25, 1.28) |
| Induction | 0.88 (0.87, 0.90) | 0.76 (0.75 0.77) | 0.76 (0.75, 0.77) | 0.77 (0.76, 0.78) |
| Vacuum delivery | 1.37 (1.35, 1.39) | 1.07 (1.05, 1.09) | 1.08 (1.06, 1.10) | 1.07 (1.05, 1.09) |
| Placental previa | 0.72 (0.68, 0.77) | 1.05 (0.99, 1.11) | 1.02 (0.97, 1.08) | 1.06 (1.00, 1.12) |
| Placental abruption | 0.83 (0.76, 0.91) | 0.95 (0.87, 1.04) | 0.94 (0.86, 1.03) | 0.96 (0.88, 1.05) |
| PPH | 0.91 (0.89, 0.92) | 0.93 (0.91, 0.95) | 0.93 (0.91, 0.94) | 0.96 (0.94, 0.97) |
| PH | 0.74 (0.63, 0.88) | 1.15 (0.98, 1.37) | 1.19 (1.00, 1.41) | 1.17 (0.98, 1.38) |
| UAE | 0.75 (0.64, 0.87) | 1.04 (0.89, 1.20) | 1.02 (0.88, 1.18) | 1.07 (0.92, 1.24) |

Abbreviations: GDM: Gestational diabetes mellitus, PPH: Postpartum hemorrhage, PH: Peripartum hysterectomy, UAE: Uterine arterial embolization

Model 1 is adjusted for age.

Model 2 is adjusted for age and parity.

Model 3 is adjusted for age, parity, and CCI

The international research collaboration Reproductive Outcomes And Migration reviewed published studies and found that immigrants in countries of resettlement have a greater risk of GDM than do women in receiving countries [10]. Similarly, we found that immigrant women had a lower prevalence but a higher risk of GDM than did Korean women after adjustment for confounding factors. Various explanations are possible for these associations. First, there was considerable heterogeneity in the prevalence of GDM in Asia, with a relatively low prevalence in Korea [17]. Therefore, foreigners may have a relatively high incidence of GDM because they show prevalence based on the characteristics of their home countries. Second, it has been reported that pre-pregnancy and pregnancy dietary patterns are associated with the development of GDM [18, 19]. Dietary acculturation is one of the cultural elements into which female immigrants first assimilate, while living away from their home country [20, 21]. Therefore, there is a possibility that the risk of GDM increased through changes in dietary pattern after moving to Korea. However, due to the lack of relevant data, further research will be needed to determine the exact cause for these differences.

GDM is associated with adverse pregnancy outcomes, including gestational hypertensive disorders, fetal macrosomia, and cesarean delivery, [22] as well as an increased risk of developing type 2 DM after delivery [23]. Therefore, there is a need for systematic and continuous management of GDM both during pregnancy and after delivery. However, it has been reported that before the diagnosis of GDM, knowledge and awareness of diabetes were low in immigrant women. Moreover, the dietary advice that was received was seen to be challenging in the context of culturally different food habits; consequently, managing diet after diagnosis proved difficult [24]. Thus, as these groups may be vulnerable to management of GDM during pregnancy and postpartum, efforts will be needed to manage them appropriately.

In this study, the cesarean section rate was lower in immigrant women than in Korean women, but after adjustment for age, parity and CCI, the odds for cesarean section was higher in immigrant women. Immigrant women were younger than Korean women therefore age might be a factor which influence the lower rate of CS before adjustment. These results are consistent with those of a previous study reporting that cesarean rates between migrants and non-migrants differed in 69% of studies [9]. Thus, it is interesting finding, and although the exact cause is unknown, the increased cesarean rates for immigrant women can probably be explained by multiple factors. First, it may be due to the characteristics of immigrant women; for example, Asian women, particularly those from Southeast Asia, are generally shorter than Korean women [25]. Indeed, it has been well reported that maternal height exerts an effect on the risk of CS, with increasing risk of CS with decreasing maternal height [26]. Moreover, the increased rates for immigrant women can probably be explained by other factors such as social and health determinants, communication barriers, and cultural preferences [9].

Consistent with results of other studies, [27] the risk of preeclampsia was lower in immigrant women than in Korean women. Moreover, the association with PPH is inconsistent across studies, with some studies reporting an increased risk in immigrants, [28, 29] while others did not [30]. In this study, the risk of PPH was lower in immigrant women, but there was no significant difference in the occurrence of PH and UAE between the two groups.

In this study, we adjusted for CCI as a covariate to correct the pre-pregnancy health status and found that immigrant women had a lower CCI score than Korean women. However, it is unclear whether these results are due to the health of the immigrant women or the lack of information on medical records in Korea after the migration. Thus, further studies are needed to evaluate pregnancy outcomes according to maternal ethnicity considering pre-pregnancy health status.

This is one of the largest population based study on pregnancy outcome of interracial marriages without missing data. However some limitations should be acknowledged when

interpreting our findings. As our study was confined to claims data on reimbursement for a medical service, we could not determine the effect on pregnancy outcomes according to the exact maternal nationality. Therefore, immigrant women may be mixed with Asians and Caucasians, and there may be differences in pregnancy outcomes among them. As such, future studies that reflect the exact nationality will be needed. Second, although the HIRA database comprises representative data of the whole country, it lacks information on uninsured medical claims, employment status, educational level as well as clinical information before the period we studied. In particular, it was difficult to obtain previous medical histories of migrant women in their previous country. Moreover, there are no reported characteristics on the women's partners. Lastly, there appears to be no missing data since we could assess to data of women who had deliveries covered by national health insurance system.

Our study with large population data demonstrated different pregnancy outcomes of immigrant women. Increased risk of GDM, and CS were showed after adjusting maternal age, parity and CCI. This result will help clinicians to consider different pregnancy outcomes when they meet immigrant women. Eventually, predicting and preparing for adverse pregnancy outcome in advance will contribute to improving pregnancy outcomes.

## Author Contributions

**Conceptualization:** Geum Joon Cho, Hyun Sun Ko, Hae Joong Cho, Seong Yeon Hong, Young Ju Jeong.

**Data curation:** Geum Joon Cho, Ho Yeon Kim, Eunjin Noh, Young Ju Jeong.

**Formal analysis:** Eunjin Noh.

**Methodology:** Geum Joon Cho, Seong Yeon Hong, Eunjin Noh, Young Ju Jeong.

**Writing – original draft:** Geum Joon Cho, Young Ju Jeong.

**Writing – review & editing:** Ho Yeon Kim, Hyun Sun Ko, Hae Joong Cho, Seong Yeon Hong.

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
