## [Decision Letter · Decision Letter 0]

3 Sep 2021

PONE-D-21-06981

Pregnancy outcomes of immigrant women living in Korea: A population-based study

PLOS ONE

Dear Dr. Jeong,

Thank you for submitting your manuscript to PLOS ONE. After careful consideration, we feel that it has merit but does not fully meet PLOS ONE’s publication criteria as it currently stands. Therefore, we invite you to submit a revised version of the manuscript that addresses the points raised during the review process.

We look forward to receiving your revised manuscript.

Kind regards,

Rubeena Zakar, Ph.D

Academic Editor

PLOS ONE

Journal Requirements:

2. PLOS requires an ORCID iD for the corresponding author in Editorial Manager on papers submitted after December 6th, 2016. Please ensure that you have an ORCID iD and that it is validated in Editorial Manager. To do this, go to ‘Update my Information’ (in the upper left-hand corner of the main menu), and click on the Fetch/Validate link next to the ORCID field. This will take you to the ORCID site and allow you to create a new iD or authenticate a pre-existing iD in Editorial Manager. Please see the following video for instructions on linking an ORCID iD to your Editorial Manager account: https://www.youtube.com/watch?v=_xcclfuvtxQ.

**3.** Your ethics statement should only appear in the Methods section of your manuscript. If your ethics statement is written in any section besides the Methods, please move it to the Methods section and delete it from any other section. Please ensure that your ethics statement is included in your manuscript, as the ethics statement entered into the online submission form will not be published alongside your manuscript.

Reviewers' comments:

Reviewer's Responses to Questions

**Comments to the Author**

1. Is the manuscript technically sound, and do the data support the conclusions?

Reviewer #1: Partly

Reviewer #2: Yes

2. Has the statistical analysis been performed appropriately and rigorously? 

Reviewer #1: Yes

Reviewer #2: Yes

3. Have the authors made all data underlying the findings in their manuscript fully available?

Reviewer #1: No

Reviewer #2: No

4. Is the manuscript presented in an intelligible fashion and written in standard English?

Reviewer #1: Yes

Reviewer #2: Yes

5. Review Comments to the Author

Reviewer #1: This article addresses an actual problem: the pregnancy outcomes among immigrant women in Korea. There are a number of issues with the methods and analysis that need to be clarified/addressed. Below are more specific comments by section:

Abstract:

Since you are calculating the OR, you should consider replacing “risk” by “odds”.

Introduction

I do not see what does the information about interracial marriages adds in the section.

The authors take a rather narrow view of data publication,

Materials & Methods / Datasets and outcomes

You are not describing from where you are taking the variable age from.

Results section

This section needs to be improved. The variables selected for describing the study population are not exhaustive. If available, authors could also present information on employment status and educational level, previous c-section, Hypertension. Diabetes, for basic characteristics and potential risk factors. If you do not have information on these previous risk factors, you might want to mention it as a limitation.

Information regarding the % of missing data on each variable (you refer to this in the discussion section) could be useful

Table 1 includes information about outcomes that shouldn’t be described here. (GBD, preeclampsia. CS, induction etc)

When describing table 2 authors should refer as odds in state of risk. You might want to include information on low birth weigh and/or preterm birth since part of your background information includes those outcomes.

Discussion section

The first paragraph is confusing. Which are you main findings?

Line 179. You are including new information (birthweight) that was not mention on the results section.

Try to stress what your data adds to the existing body of evidence

Reviewer #2: The authors made a great presentation of the study findings - clearly linking the problem, aims, research questions, methodology, analysis results/findings, discussions, conclusions, and recommendations. Also, study clearly demonstrated associating factors with pregnancy outcomes in both groups. However, authors could provide more details in explanation to help the audience understand how caesarean section rate affects pregnancy outcome among immigrant women and state which specific cofounding factors affected (increased) caesarean section among immigrant women [line 176 & 177].

Over all, I think this is a technically sound and well documented study with details clearly stated findings and recommendations. Authors should state in the conclusion who (government, health workers, policy makers,…) is responsible for implementing effective pregnancy and postpartum management change/intervention for immigrant women [line 206 & 207] to enhance social change.

6. PLOS authors have the option to publish the peer review history of their article (what does this mean?). If published, this will include your full peer review and any attached files.

Reviewer #1: No

Reviewer #2: **Yes: **Gabriel Adebe PhD

---

## [Author Response · Author response to Decision Letter 0]

23 Dec 2021

Reviewer #1: This article addresses an actual problem: the pregnancy outcomes among immigrant women in Korea. There are a number of issues with the methods and analysis that need to be clarified/addressed. Below are more specific comments by section:

Abstract:

Since you are calculating the OR, you should consider replacing “risk” by “odds”.

Response: As you have pointed out, we wrote “odds” instead of “risk” as suggested. 

Introduction

I do not see what does the information about interracial marriages adds in the section.

The authors take a rather narrow view of data publication,

Response: We fully agree with your suggestion. We corrected the introduction as follows (Line 65-75)

Preterm birth and low birth weight are serious adverse pregnancy outcomes. They are not only related to infant morbidity and mortality but also increase risk for chronic diseases such as cardiovascular diseases and metabolic syndrome during lifetime[1]. Several biological risk factors have been identified such as maternal age, body mass index and smoking and alcohol consumption. Further influences of racial and ethnic disparities have been documented. Although, many obstetric investigations across different countries have limited adjustment for ethnicity, compared to native women, immigrant women have an increased risk of preterm birth, low birth weight and cesarean section.[1,2, 4,5] It has been reported that immigrants in countries of resettlement have a greater risk of gestational diabetes mellitus (GDM) than women in receiving countries.[3] Thus, influxes of migrant women of childbearing age to receiving countries have made their perinatal health status a key priority for many governments.[5]

Materials & Methods / Datasets and outcomes

You are not describing from where you are taking the variable age from.

Response: We added “HIRA” in this sentence (Line 94-95) as following: “The HIRA database we received included unidentifiable codes that represented individuals, together with age, diagnosis, and a list of prescribed procedures.” 

Results section

This section needs to be improved. The variables selected for describing the study population are not exhaustive. If available, authors could also present information on employment status and educational level, previous c-section, Hypertension. Diabetes, for basic characteristics and potential risk factors. If you do not have information on these previous risk factors, you might want to mention it as a limitation.

Response: We included limited information obtained from HIRA database and wrote limitation of certain risk factors (Line 201-205) in discussion. “Second, although the HIRA database comprises representative data of the whole country, it lacks information on uninsured medical claims, employment status, educational level as well as clinical information before the period we studied. In particular, migrant women do not know their past history in their previous country. ”

Information regarding the % of missing data on each variable (you refer to this in the discussion section) could be useful

Response: The strength of our study is that there is no missing data in HIRA database. We added this comment on page 13, line 206-207.

Table 1 includes information about outcomes that shouldn’t be described here. (GBD, preeclampsia. CS, induction etc)

Response: We reorganized table 1 into table 1 which includes basic characteristics and table 2 which includes pregnancy outcome. 

When describing table 2 authors should refer as odds in state of risk. You might want to include information on low birth weigh and/or preterm birth since part of your background information includes those outcomes.

Response: We changed the risk to odds in Line 134-142. 

Discussion section

The first paragraph is confusing. Which are you main findings?

Response: We clarified the first paragraph of discussion as follows. 

“Immigrant women had a lower prevalence of preeclampsia and PPH and higher prevalence of GDM, CS and placenta previa compared to Korean women after adjustment for age, parity, and CCI.” 

Line 179. You are including new information (birthweight) that was not mention on the results section.

Response: We deleted this irrelevant sentence. “In line with this, we previously reported that the birthweight was lower for the children of foreign women than for those of Korean women.”

Try to stress what your data adds to the existing body of evidence

Response: Our study with large population data demonstrated different pregnancy outcomes of immigrant women. Increased risk of GDM, CS and placenta previa were showed after adjusting maternal age, parity and CCI. This result will help clinicians to consider different pregnancy outcomes when they meet immigrant women. Eventually, predicting and preparing for adverse pregnancy outcome in advance will contribute to improving pregnancy outcomes. 

 

Reviewer #2: The authors made a great presentation of the study findings - clearly linking the problem, aims, research questions, methodology, analysis results/findings, discussions, conclusions, and recommendations. Also, study clearly demonstrated associating factors with pregnancy outcomes in both groups. However, authors could provide more details in explanation to help the audience understand how caesarean section rate affects pregnancy outcome among immigrant women and state which specific cofounding factors affected (increased) caesarean section among immigrant women [line 176 & 177].

Response: We added following statement in Line 176-179. “In this study, the cesarean section rate was lower in immigrant women than in Korean women, but after adjustment for age, parity and CCI, the risk for cesarean section was higher in immigrant women. Immigrant women were younger than Korean women therefore age might be a factor which influences the lower rate of CS before adjustment.”

Over all, I think this is a technically sound and well documented study with details clearly stated findings and recommendations. Authors should state in the conclusion who (government, health workers, policy makers,…) is responsible for implementing effective pregnancy and postpartum management change/intervention for immigrant women [line 206 & 207] to enhance social change.

Response: We added following statement in the last paragraph of this manuscript(Line 207-209). “These characteristics should be reflected in Korean medical system to enhance social changes and effective pregnancy and postpartum management by healthcare providers will be necessary for immigrant women.”

---

## [Decision Letter · Decision Letter 1]

28 Mar 2022

PONE-D-21-06981R1Pregnancy outcomes of immigrant women living in Korea: A population-based studyPLOS ONE

Dear Dr. Ju Jeong

Thank you for submitting your manuscript to PLOS ONE. After careful consideration, we feel that it has merit but does not fully meet PLOS ONE’s publication criteria as it currently stands. Therefore, we invite you to submit a revised version of the manuscript that addresses the points raised during the review process.

We look forward to receiving your revised manuscript.

Kind regards,

Rubeena Zakar, Ph.D

Academic Editor

PLOS ONE

Journal Requirements:

Reviewers' comments:

Reviewer's Responses to Questions

**Comments to the Author**

1. If the authors have adequately addressed your comments raised in a previous round of review and you feel that this manuscript is now acceptable for publication, you may indicate that here to bypass the “Comments to the Author” section, enter your conflict of interest statement in the “Confidential to Editor” section, and submit your "Accept" recommendation.

Reviewer #1: (No Response)

Reviewer #2: All comments have been addressed

2. Is the manuscript technically sound, and do the data support the conclusions?

Reviewer #1: Partly

Reviewer #2: Yes

3. Has the statistical analysis been performed appropriately and rigorously? 

Reviewer #1: Yes

Reviewer #2: Yes

4. Have the authors made all data underlying the findings in their manuscript fully available?

Reviewer #1: Yes

Reviewer #2: Yes

5. Is the manuscript presented in an intelligible fashion and written in standard English?

Reviewer #1: No

Reviewer #2: Yes

6. Review Comments to the Author

Reviewer #1: The paper needs language editing. Consider a language revision.

When explaining The Charlson Comorbidity Index it would be useful to explain if the higher the score the more comorbid conditions are present (line 110)

From line 135 to 140 you described table 1, but in the middle of the paragraph there is information related to table 2 (The cesarean section (CS) rate was lower in foreign women than Korean women. The incidence of placental abruption, placental previa, PPH, PH, and UAE

were lower in immigrant women than Korean women.) that information was repeated in line 150.

Table 2. you included: Values are expressed as mean (SD) or %. But there are not values expressed as mean and SD

Table 3. should be named: Odds ratio and 95% confidence intervals of adverse pregnancy out 155 comes of immigrant women instead of Odds ratio and 95% confidence intervals of the risk of adverse pregnancy out 155 comes of immigrant women

This sentence is confusing:

In particular, migrant women do not know their past history in their previous country

Which is their past history they don’t know?

In table 1 it says “Primparity” check spelling

Line 187: it says risks, it should say odds

190. I still see the sentences: In line with this, we previously reported that the birthweight was lower for the children of foreign women than for those of Korean women.[11]

In this sense: I asked you before, Try to stress what your data adds to the existing body of evidence

Your Response was: “Our study with large population data demonstrated different pregnancy

outcomes of immigrant women. Increased risk of GDM, CS and placenta previa were

showed after adjusting maternal age, parity and CCI. This result will help clinicians to

consider different pregnancy outcomes when they meet immigrant women. Eventually,

predicting and preparing for adverse pregnancy outcome in advance will contribute to

improving pregnancy outcomes.”

That information was not included in the discussion section. Besides, there was not an increased risk of placenta previa (the confidence interval starts in1)

I think you should do emphasis on the results presented in table 3.: After adjusting by age, parity and CCI, which are the variables you are expected to adjust by, (considering there were differences between the 2 groups in table 1). Findings are that: to be immigrant increase the risk of having GDM, and CS. But protects against having preeclampsia, and induction.

Reviewer #2: Earlier concerns addressed, paper appears consistent with sections observed with minor changes. I believe the paper is publish-ready

7. PLOS authors have the option to publish the peer review history of their article (what does this mean?). If published, this will include your full peer review and any attached files.

Reviewer #1: No

Reviewer #2: **Yes: **Adebe Gabriel Aondofa (Ph.D, MPH)

---

## [Author Response · Author response to Decision Letter 1]

25 Apr 2022

Reviewer #1: The paper needs language editing. Consider a language revision.

: We finished language editing as suggested. 

When explaining The Charlson Comorbidity Index it would be useful to explain if the higher the score the more comorbid conditions are present (line 110)

: We added this statement in line 112 “that the higher the score the more comorbid conditions are present’

From line 135 to 140 you described table 1, but in the middle of the paragraph there is information related to table 2 (The cesarean section (CS) rate was lower in foreign women than Korean women. The incidence of placental abruption, placental previa, PPH, PH, and UAE

were lower in immigrant women than Korean women.) that information was repeated in line 150.

:We deleted this sentence in line 130. 

Table 2. you included: Values are expressed as mean (SD) or %. But there are not values expressed as mean and SD

: We wrote “values are expressed as number, %.”

Table 3. should be named: Odds ratio and 95% confidence intervals of adverse pregnancy out 155 comes of immigrant women instead of Odds ratio and 95% confidence intervals of the risk of adverse pregnancy out 155 comes of immigrant women

: We corrected as suggested. 

This sentence is confusing:

In particular, migrant women do not know their past history in their previous country

Which is their past history they don’t know?

: We corrected this sentence as follows; In particular it was difficult to obtain previous medical histories of migrant women in their previous country. 

In table 1 it says “Primparity” check spelling

: We corrected as suggested

Line 187: it says risks, it should say odds 190.

: We corrected as suggested

I still see the sentences: In line with this, we previously reported that the birthweight was lower for the children of foreign women than for those of Korean women.[11]

: We deleted this sentence as suggested. 

In this sense: I asked you before, Try to stress what your data adds to the existing body of evidence

Your Response was: “Our study with large population data demonstrated different pregnancy

outcomes of immigrant women. Increased risk of GDM, CS and placenta previa were

showed after adjusting maternal age, parity and CCI. This result will help clinicians to

consider different pregnancy outcomes when they meet immigrant women. Eventually,

predicting and preparing for adverse pregnancy outcome in advance will contribute to

improving pregnancy outcomes.”

That information was not included in the discussion section. Besides, there was not an increased risk of placenta previa (the confidence interval starts in1)

: We added this sentences in discussion section. The risk of placenta previa was corrected in abstract, results and discussion. 

I think you should do emphasis on the results presented in table 3.: After adjusting by age, parity and CCI, which are the variables you are expected to adjust by, (considering there were differences between the 2 groups in table 1). Findings are that: to be immigrant increase the risk of having GDM, and CS. But protects against having preeclampsia, and induction.

: We added following statement in the first paragraph of discussion (Line 166-168)

Immigrant women were protected against having preeclampsia and PPH and higher risk of GDM and CS compared to Korean women after adjustment for age, parity and CCI.

---

## [Decision Letter · Decision Letter 2]

10 Jul 2022

PONE-D-21-06981R2Pregnancy outcomes of immigrant women living in Korea: A population-based studyPLOS ONE

Dear Dr.  Ju Jeong,

Thank you for submitting your manuscript to PLOS ONE. After careful consideration, we feel that it has merit but does not fully meet PLOS ONE’s publication criteria as it currently stands. Therefore, we invite you to submit a revised version of the manuscript that addresses the points raised during the review process.

We look forward to receiving your revised manuscript.

Kind regards,

Rubeena Zakar, Ph.D

Section Editor

PLOS ONE

Reviewers' comments:

Reviewer's Responses to Questions

**Comments to the Author**

1. If the authors have adequately addressed your comments raised in a previous round of review and you feel that this manuscript is now acceptable for publication, you may indicate that here to bypass the “Comments to the Author” section, enter your conflict of interest statement in the “Confidential to Editor” section, and submit your "Accept" recommendation.

Reviewer #1: All comments have been addressed

Reviewer #3: (No Response)

Reviewer #4: (No Response)

2. Is the manuscript technically sound, and do the data support the conclusions?

Reviewer #1: Yes

Reviewer #3: Yes

Reviewer #4: Partly

3. Has the statistical analysis been performed appropriately and rigorously? 

Reviewer #1: Yes

Reviewer #3: Yes

Reviewer #4: No

4. Have the authors made all data underlying the findings in their manuscript fully available?

Reviewer #1: Yes

Reviewer #3: No

Reviewer #4: Yes

5. Is the manuscript presented in an intelligible fashion and written in standard English?

Reviewer #1: Yes

Reviewer #3: Yes

Reviewer #4: Yes

6. Review Comments to the Author

Reviewer #1: All my comments were addressed.

Well done.

I have no further comments.

...........................................................

Reviewer #3: The authors have done well in providing evidence on pregnancy outcomes among immigrants in Korea. Authors could consider the following issues to strengthen the paper.

Introduction

Authors provide a very brief introduction to the topic of interest and seem to miss out important detail. First, about half of the first paragraph focuses on preterm birth and low birth weight and their predictors when the study itself (including analysis) rarely considers these outcomes. Second, the detail on interracial marriage in Korea is great to provide information on the immigration status in Korea. However, the argument is too one-sided and does not feed into the issues raised in the subsequent sentences. I think a reviewer in the earlier revisions of this work raised this issue. Authors could strengthen their introduction by considering the following:

1. What pregnancy outcomes are highly prevalent in Korea?

2. Are there studies that report on these outcomes by immigration status in Korea? Kindly report on this in addition to what has been done in other countries.

3. Justify why immigrants in Korea should be focused.

Methods

1. Authors should provide justification for the duration 2007 - 2016.

2. Can authors clearly define their outcome variables? For instance a number of variables are mentioned and this includes parity and multiple pregnancies which i presume are not the main outcome variables of interest they seek to look at.

3. Regarding section titled "covariates", authors seem to focus mainly on ICC and how it is created from pre-pregnancy factors but there is no information on the specific factors that were combined to create this index. Neither is there information on how this index is created. Is this something that exist in the database or it is an index authors created themselves. Authors should kindly provide more detail regarding this.

4. Since the data for the study covers the period between 2007 and 2016, there is the possibility of multiple pregnancies per woman. How do authors account for this?

5. Line 118: can authors specify the exact group they are referring to?

6. Authors also do not mention the final sample size they used.

7. I think authors should dedicate a section for ethical consideration to outline all forms of ethics that apply to their work; data availability and how it was accessed.

Results

1. lines 131-133: results is not in Table 1. Authors should check this and interpret the other variables in table one in place of this.

Discussion

I find the results to be well discussed and the limitations well laid out.

Reviewer #4: I appreciate the opportunity to review your manuscript. Please find below some important areas in which the manuscript can be clarified or improved from the statistical perspective.

1) The methods section of the manuscript can be better organized by using sub headings e.g. study design and data source, study outcomes, other measures, etc. Please follow the CONSORT guidelines or any other required by the journal.

2) Primary and secondary outcomes must be clearly identified and distinguished in the abstract, methods section, statistical methods, results, and discussion.

3) Please mention all the covariates that were examined in the covariates section of the manuscript.

4) In the statistical analysis section, please include that model was run stepwise and reason behind running 3 separate models.

5) In all the tables in which statistical tests were conducted, the footnotes need to specify which test was used to derive each reported P-value.

6) For age please clarify in the tables if it is mean age or median.

7) Authors have multiple outcome variables but use 0.05 as significance criterion throughout. Please use Bonferroni correction or other method to adjust for multiple testing. Otherwise, the results will have an elevated chance of being false positives.

8) Was there any missing data in your study? How was it handled?

7. PLOS authors have the option to publish the peer review history of their article (what does this mean?). If published, this will include your full peer review and any attached files.

Reviewer #1: No

Reviewer #3: No

Reviewer #4: No

---

## [Author Response · Author response to Decision Letter 2]

8 Sep 2022

Response to reviewers

Reviewer #3: The authors have done well in providing evidence on pregnancy outcomes among immigrants in Korea. Authors could consider the following issues to strengthen the paper.

Introduction

Authors provide a very brief introduction to the topic of interest and seem to miss out important detail. First, about half of the first paragraph focuses on preterm birth and low birth weight and their predictors when the study itself (including analysis) rarely considers these outcomes. Second, the detail on interracial marriage in Korea is great to provide information on the immigration status in Korea. However, the argument is too one-sided and does not feed into the issues raised in the subsequent sentences. I think a reviewer in the earlier revisions of this work raised this issue. Authors could strengthen their introduction by considering the following:

1. What pregnancy outcomes are highly prevalent in Korea?

In South Korea, the incidence of GDM, hypertensive disorder in pregnancy, and preterm birth and the rate of cesarean section have been steadily increasing. GDM accounts for 15.8% in 2017 to 18.01% in 2021, preeclampsia has doubled 2.75% in 2017 to 5.34% in 2021, preterm birth rate was 5.8% in 2010 and 8.5% in 2020 

2. Are there studies that report on these outcomes by immigration status in Korea? Kindly report on this in addition to what has been done in other countries.

3. Justify why immigrants in Korea should be focused.

We appreciate above suggestions mainly focusing on why we need to study immigrant pregnant women and their association with adverse pregnancy outcomes. Therefore we rewrote and organized introduction.

Methods

1. Authors should pro vide justification for the duration 2007 - 2016.

At the time of planning the study, period between 2007 and 2016 could be accessed and analyzed. In the future, it is also necessary to use this data to see long-term changes.

2. Can authors clearly define their outcome variables? For instance a number of variables are mentioned and this includes parity and multiple pregnancies which i presume are not the main outcome variables of interest they seek to look at.

We defined variables we looked at as suggested in methods

Parity and the rate of cesarean section, induction, vacuum delivery, uterine arterial embolization (UAE), and peripartum hysterectomy (PH) were also identified using the presence of a Korea Medical Insurance electronic data interchange (EDI) code.

3. Regarding section titled "covariates", authors seem to focus mainly on ICC and how it is created from pre-pregnancy factors but there is no information on the specific factors that were combined to create this index. Neither is there information on how this index is created. Is this something that exist in the database or it is an index authors created themselves. Authors should kindly provide more detail regarding this.

CCI was used in this manuscript to represent comorbid status of patients before pregnancy as a simple and widely used method to identify underlying diseases. We added more detail and reference on CCI in the methods. 

The CCI has been known to be a useful and most widely used tool to measure comorbid disease status including cardiovascular diseases, diabetes, malignancies and autoimmune diseases or casemix in health care databases that the higher the score the more comorbid conditions are present. Acute myocardial infarction, Congestive heart failure, Peripheral vascular disease, Cerebral vascular accident, Dementia, Pulmonary disease, Connective tissue disorder, Peptic ulcer, Liver disease, Diabetes, Diabetes complications, Paraplegia, Renal disease, Cancer, Metastatic cancer, Severe liver disease, HIV were analyzed in this study.

4. Since the data for the study covers the period between 2007 and 2016, there is the possibility of multiple pregnancies per woman. How do authors account for this?

Multiple pregnancies were included in this data set but we did not put as a variable. We also deleted “multiple pregnancy” in method section. 

5. Line 118: can authors specify the exact group they are referring to?

We added following sentence to define group 

(Line 130-131) The Student’s t-test was used to compare continuous variables between Korean and immigrant pregnant women

6. Authors also do not mention the final sample size they used.

Final sample size was 4,439,778. 

7. I think authors should dedicate a section for ethical consideration to outline all forms of ethics that apply to their work; data availability and how it was accessed.

We added following statement in method. 

Ethical consideration

This study was approved by the Institutional Review Committees of Korea University Guro Hospital (KUGH17273). The informed consent was waived by this IRB because the HIRA database does not include individual identities and because of retrospective nature of this study. This study comprises of third-party data therefore authors cannot share data nor legally distribute. 

Results

1. lines 131-133: results is not in Table 1. Authors should check this and interpret the other variables in table one in place of this.

We corrected result of table 1 in manuscript. 

Table 1 shows the basic characteristics of the study population according to maternal ethnicity. Foreign women were younger and more primiparous than Korean women and tended to have lower CCI score than Korean women.

Discussion

I find the results to be well discussed and the limitations well laid out.

We appreciate this comment. 

Reviewer #4: I appreciate the opportunity to review your manuscript. Please find below some important areas in which the manuscript can be clarified or improved from the statistical perspective.

1) The methods section of the manuscript can be better organized by using sub headings e.g. study design and data source, study outcomes, other measures, etc. Please follow the CONSORT guidelines or any other required by the journal.

We used following headings in methods. 

Study desing-data characteristics-Ethical consideration-Dataset and outcomes-Covariates-Statistical analysis 

2) Primary and secondary outcomes must be clearly identified and distinguished in the abstract, methods section, statistical methods, results, and discussion.

We used ‘main outcome’ and added following sentence in the abstract, method, results and discussion. 

Main outcome of this study were adverse pregnancy outcomes including GDM, preeclampsia, cesarean section, induction of labor, vacuum delivery, placenta previa, placenta abruption, PPH, peripartum hysterectomy and UAE. Age, parity and CCI were adjusted for multiple logistic regression analysis.

3) Please mention all the covariates that were examined in the covariates section of the manuscript.

CCI used in this study included following diseases. 

Acute myocardial infarction, Congestive heart failure, Peripheral vascular disease, Cerebral vascular accident, Dementia, Pulmonary disease, Connective tissue disorder, Peptic ulcer, Liver disease, Diabetes, Diabetes complications, Paraplegia, Renal disease, Cancer, Metastatic cancer, Severe liver disease, HIV were analyzed in this study.

4) In the statistical analysis section, please include that model was run stepwise and reason behind running 3 separate models.

We added ‘the model was run stepwise’ in method. The running 3 separate models was conducted to identify the variables that affect the most. 

5) In all the tables in which statistical tests were conducted, the footnotes need to specify which test was used to derive each reported P-value.

We added all the statistical tests used in the footnotes of each tables. 

6) For age please clarify in the tables if it is mean age or median.

For age, the results indicate mean±standard deviation. 

7) Authors have multiple outcome variables but use 0.05 as significance criterion throughout. Please use Bonferroni correction or other method to adjust for multiple testing. Otherwise, the results will have an elevated chance of being false positives.

We appreciate thorough statistical suggestions. In this study, multiple outcome variables were analyzed. Although each variable may be related to each other, each variable is an important subject identified in obstetrics. Therefore, age, parity, and CCI were adjusted in regression model.

8) Was there any missing data in your study? How was it handled?

This study is based on the claim data and since the diagnosis was confirmed using the icd-10 code, there is no missing data among the mothers charged for delivery. We added this issue in limitation of discussion.

---

## [Decision Letter · Decision Letter 3]

14 Nov 2022

Pregnancy outcomes of immigrant women living in Korea: A population-based study

PONE-D-21-06981R3

Dear Dr. Jeong,

We’re pleased to inform you that your manuscript has been judged scientifically suitable for publication and will be formally accepted for publication once it meets all outstanding technical requirements.

Kind regards,

Rubeena Zakar, Ph.D

Section Editor

PLOS ONE

Additional Editor Comments (optional):

Reviewers' comments:

Reviewer's Responses to Questions

**Comments to the Author**

1. If the authors have adequately addressed your comments raised in a previous round of review and you feel that this manuscript is now acceptable for publication, you may indicate that here to bypass the “Comments to the Author” section, enter your conflict of interest statement in the “Confidential to Editor” section, and submit your "Accept" recommendation.

Reviewer #1: All comments have been addressed

Reviewer #5: All comments have been addressed

Reviewer #6: (No Response)

Reviewer #7: All comments have been addressed

2. Is the manuscript technically sound, and do the data support the conclusions?

Reviewer #1: Yes

Reviewer #5: Yes

Reviewer #6: Yes

Reviewer #7: Yes

3. Has the statistical analysis been performed appropriately and rigorously? 

Reviewer #1: Yes

Reviewer #5: Yes

Reviewer #6: Yes

Reviewer #7: Yes

4. Have the authors made all data underlying the findings in their manuscript fully available?

Reviewer #1: Yes

Reviewer #5: Yes

Reviewer #6: Yes

Reviewer #7: Yes

5. Is the manuscript presented in an intelligible fashion and written in standard English?

Reviewer #1: No

Reviewer #5: Yes

Reviewer #6: Yes

Reviewer #7: Yes

6. Review Comments to the Author

Reviewer #1: In the introduction section from line 88 to line 90 there are empty lines.

Line 147 is should be resultS

Table 2. should be named: “pregnancy outcomes of the study population” without “The”

Table 3. should be named: Odds ratio and 95% confidence intervals of adverse pregnancy outcomes of immigrant women. Check out please you included an extra “155”. It was a cut & paste mistake for sure.

Line 198. Table 3 does not “represents”, I guess you wanted to say “presents”

Also in line 198, I would add the word “unadjusted” in this sentence: “The odds of GDM, preeclampsia, CS, placental abruption, placental previa, PPH, PH, and UAE was lower in foreign women than Korean women.”

In line 215 and 216 why you did not mention the protection against induction?

“Immigrant women were protected against having preeclampsia and PPH and

216 higher risk of GDM, and CS compared to Korean women after adjustment for age, parity, and CCI.”

Line 261 it should be outcomeS

There are still some spelling mistakes. I suggest you perform a language edition to your manuscript.

Reviewer #5: None-------------------------------------------------------------------------------------------------------------------------------------------

Reviewer #6: Appreciate the work of author. Author has clearly mentioned the study population, just a simple question to author about the rational behind selection of 9 years study duration? Is it just because the medical record about pregnancy during that time were available or there is any rational for interracial marriage and pregnancies during that period?

Thank you

Reviewer #7: The title of the study, aim and basic statistical analysis demonstrating pregnancy outcomes between Korean women and immigrant women appear appropriate and linked adequately.

7. PLOS authors have the option to publish the peer review history of their article (what does this mean?). If published, this will include your full peer review and any attached files.

Reviewer #1: No

Reviewer #5: No

Reviewer #6: No

Reviewer #7: **Yes: **Nnodimele Onuigbo ATULOMAH PhD

---

## [Editor Report · Acceptance letter]

17 Nov 2022

PONE-D-21-06981R3 

Pregnancy outcomes of immigrant women living in Korea: A population-based study 

Dear Dr. Jeong:

I'm pleased to inform you that your manuscript has been deemed suitable for publication in PLOS ONE. Congratulations! Your manuscript is now with our production department. 

Kind regards, 

on behalf of

Dr. Rubeena Zakar 

Section Editor

PLOS ONE